# *KDM6B* Variants May Contribute to the Pathophysiology of Human Cerebral Folate Deficiency

**DOI:** 10.3390/biology12010074

**Published:** 2022-12-31

**Authors:** Xiao Han, Xuanye Cao, Robert M. Cabrera, Paula Andrea Pimienta Ramirez, Cuilian Zhang, Vincent T. Ramaekers, Richard H. Finnell, Yunping Lei

**Affiliations:** 1Department of Reproductive Medicine Center, Henan Provincial People’s Hospital, People’s Hospital of Zhengzhou University, Zhengzhou 450003, China; 2Center for Precision Environmental Health, Department of Molecular and Cellular Biology, Baylor College of Medicine, Houston, TX 77030, USA; 3Department of Pediatric Neurology, University Hospital Center Liège, 4000 Liège, Belgium; 4Departments of Molecular and Human Genetics and Medicine, Baylor College of Medicine, Houston, TX 77030, USA

**Keywords:** KDM6B, cerebral folate deficiency, FOLR1, H3K27me2, H3K27Ac

## Abstract

**Simple Summary:**

Cerebral folate deficiency syndrome (CFD) was defined as any neurological condition that was associated with low concentrations of 5-methyltetrahydrofolate in the cerebrospinal fluid. Previous clinical studies have suggested that mutations in the folate receptor alpha (*FOLR1*) gene contribute to CFD. In this study, we identified six genetic variants in histone lysine demethylase 6B (*KDM6B*) in 48 CFD cases. We demonstrated that these *KDM6B* variants decreased FOLR1 protein expression by manipulating epigenetic markers regulating chromatin organization and gene expression. In addition, FOLR1 autoantibodies were identified in CFD patients’ serum. To the best of our knowledge, this is the first study to report that KDM6B may be a novel CFD candidate gene in humans.

**Abstract:**

(1) Background: The genetic etiology of most patients with cerebral folate deficiency (CFD) remains poorly understood. *KDM6B* variants were reported to cause neurodevelopmental diseases; however, the association between *KDM6B* and CFD is unknown; (2) Methods: Exome sequencing (ES) was performed on 48 isolated CFD cases. The effect of *KDM6B* variants on KDM6B protein expression, Histone H3 lysine 27 epigenetic modification and FOLR1 expression were examined in vitro. For each patient, serum FOLR1 autoantibodies were measured; (3) Results: Six *KDM6B* variants were identified in five CFD patients, which accounts for 10% of our CFD cohort cases. Functional experiments indicated that these *KDM6B* variants decreased the amount of KDM6B protein, which resulted in elevated H3K27me2, lower H3K27Ac and decreased FOLR1 protein concentrations. In addition, FOLR1 autoantibodies have been identified in serum; (4) Conclusion: Our study raises the possibility that *KDM6B* may be a novel CFD candidate gene in humans. Variants in *KDM6B* could downregulate FOLR1 gene expression, and might also predispose carriers to the development of FOLR1 autoantibodies.

## 1. Introduction

Cerebral folate deficiency (CFD, OMIM#: 613068) syndrome was defined as any neurological condition that was associated with low concentrations of 5-methyltetrahydrofolate (5MTHF) in the cerebrospinal fluid (CSF) [1,2]. CFD patients exhibit a wide clinical presentation, with reported symptoms starting at 4 months of age with irritability and sleep disturbances, which are often followed by psychomotor retardation, dyskinesia, cerebellar ataxia and spastic diplegia. Other symptoms include deceleration of head growth, visual disturbances and sensorineural hearing loss [3].

In humans, folate concentrations are from 1.5 to 3-fold higher in CSF than in serum, and the ratio is much higher in infancy and decreases through adolescence [4,5]. Folate is essential for normal embryonic and individual development and, especially in the central nervous system, prenatal and postnatal folate deficiencies are believed to contribute to a variety of neurological conditions, including intellectual disability, epilepsy, ataxia and pyramidal tract signs [6,7]. The most widely used treatment to correct identified deficiencies is calcium folinate or folinic acid, which has proven to be somewhat effective in alleviating many of the clinical symptoms and improve the prognosis for a better developmental and neurological outcome [8].

Several mechanisms have been demonstrated to be involved in CFD development [3]. Secondary folate deficiencies could be caused by dietary folate insufficiency, exposure to anti-folate drugs, hepatic failure or celiac disease [9]. Some rare genetic diseases are also etiologically linked to CFD, such as hereditary folate malabsorption due to *SLC46A1* (also called PCFT or HCP. OMIM#: 611672) gene variants [10] and dihydrofolate reductase deficiency (*DHFR*, OMIM#: 126060). Furthermore, it has been widely accepted that variants of the folate transport gene *FOLR1* (OMIM#: 136430) contribute to CFD etiology [11,12]. Usually, the causes of mild-to-moderate CFD in children are attributed to conditions secondary to several mitochondrial disorders, which are frequently found in children diagnosed with various organ dysfunctions, including neurological disorders. However, in most patients with moderate to severe infantile-onset CFD, the etiology has been associated with high titers of serum folate receptor-alpha autoantibodies [7,13]. In these cases, impaired folate transport across the choroid plexus causes the folate deficiencies observed in the CSF and neural tissues and contributes to the adverse developmental phenotypes [1,14].

Lysine demethylase 6B (*KDM6B*, OMIM#: 611577), is a lysine-specific demethylase that specifically demethylates di- or tri-methylated lysine 27 of histone H3 (H3K27me2 or H3K27me3) [15]. These H3K27 methylations act as a repressive epigenetic marker regulating chromatin organization and gene expression [16]. Histone methylation plays an important role in several diverse biological processes. KDM6B contributes to inflammatory responses as well as to various biological processes, including stress-induced cell senescence, development, and differentiation [17]. In 2019, Stolerman et al. [18]) reported 12 patients with de novo *KDM6B* deleterious variants presenting with neurodevelopmental delays in speech and motor development, dysmorphic facial features including a prominent nasal bridge or nose, coarse features, and widened hands and syndactyly. However, whether any of these patients had CFD was unknown.

Herein, to further explore the genetic etiology and mechanisms of human CFD, we analyzed exome data on 48 CFD cases and identified six rare missense *KDM6B* variants. We also performed functional analysis which demonstrated that *KDM6B* missense variants identified in CFD patients downregulated protein levels of FOLR1 in HeLa cell lines. Further experiments indicated that mutated KDM6B might regulate FOLR1 levels through increased H3K27me2 and decreased H3K27Ac expression, which could potentially affect FOLR1 transcription. Overall, our findings demonstrated that *KDM6B* variants contribute to human CFD for the first time, which enhanced our understanding of the etiology of this newly recognized clinical entity. 

## 2. Materials and Methods

### 2.1. Ethical Compliance 

This study was approved by the Institutional Review Board (IRB) Committee at Baylor College of Medicine (approve #: H-49549) and the Ethics Committee at Liège University Hospital approved the study with Protocol number FOL040113, registered at the Belgian number B707201316427 (FAMHP: Federal Agency for Medicines and Health Products). Informed consent form was signed by parents of all the cases included.

### 2.2. Human Subject 

Genomic DNA samples were collected from 48 CFD cases whose clinical findings were previously described in our publication [19]. Folate concentrations in cerebrospinal fluid were measured at the treating hospitals. *KDM6B* (NM_001080424.2) sequencing data from gnomAD database was used as control.

### 2.3. Sequencing Analysis 

Exome sequencing (ES) and data analysis were described previously [6]. Briefly, DNA sequencing libraires were built using NEB Next Ultra DNA Library Prep Kit (New England Biolabs, Ipswich, MA, USA). The exome regions were captured using Agilent SureSelect Human. All Exon V5 (Agilent Technologies, Santa Clara, CA, USA). Libraries were sequenced on Illumina Hi-Seq 2000 platform (Illumina, San Diego, CA, USA). Exome sequencing data in FASTQ format were mapped to hg19 using BWA alignment software, sorted and indexed by SAMtools, base recalibrated by GATK. Variants were called using GATK HaplotypeCaller following GATK Best Practice protocol. Rare deleterious were grouped by gene. *KDM6B* is ranked the first for number of rare deleterious variants in the 48 CFD cases.

### 2.4. Plasmids

pcDNA3.1+/C-(K)DYK vector and pcDNA3.1+/C-(K)DYK-KDM6B vector was purchased from Genscript. KDM6B mutant plasmids were produced by the Genescript company based on the pcDNA3.1+/C-(K)DYK-KDM6B wildtype vector.

### 2.5. Cell Culture and Transfection 

HeLa cells were cultured in Dulbecco’s Modified Eagles Medium (DMEM, Sigma, St. Louis, MO, D6429, USA), supplemented with 10% heat-inactivated fetal bovine serum (FBS, Gibco, 26140079) and 1X Antibiotic-Antimycotic (ThermoFisher Scientific, Waltham, MA, 15240-062, USA). Cell cultures were maintained at 37 °C in a humidified atmosphere incubation containing 5% CO_2_. When 50% confluency is reached, transfection was carried out using Lipofectamine 2000 (ThermoFisher Scientific, Waltham, MA, 11668019, USA) according to the manufacturer’s protocol. Medium was replaced with the culture medium without antibiotics prior to transfection. Further experiments were performed 48 h after transfection.

### 2.6. Immunofluorescence 

Immunofluorescence staining was performed to identify the localization of wildtype and mutated KDM6B protein within cells. HeLa cells were plated onto 35 mm glass-bottom dishes at a density of 0.8 × 10^6^ cells per dish and were transfected with 2 ug wildtype or mutated KDM6B plasmids when 50% confluency is reached on the second day. 48 h after transfection, cells were fixed with 4% paraformaldehyde for 30 min and cell membranns were permeabilized with 0.3% Triton X-100 in TBST for 20 min, and then blocked with 10% normal goat serum (NGS) in Tris-buffered saline with 0.1% Tween^®^ 20 Detergent (TBST) for 1 h. Cells were then incubated with FOLR1 antibody (1:100, Proteintech, Rosemont, IL, USA, 23355-1-AP) and flag antibody (1:1000, ThermoFisher Scientific, MA1-91878) overnight at 4 °C. After incubation, cells were rinsed with Phosphate-buffered saline (PBS) and incubated with secondary antibodies (1:1000, Cell Signaling Technology, Danvers, MA, USA, 8889S; 4408S) for 1 h, avoiding light exposure. Anti-fade mountant with DAPI (Invitrogen, Waltham, MA, USA, P36931) was used to stain nucleus and to protect fluorescent protein from fading. Deconvolution microscope (Nikon T2) was used for imaging.

### 2.7. Western Blotting Assay 

HeLa cells were plated in 6-well plate at a density of 0.8 × 10^6^ cells per well and were transfected with 2 ug wildtype and mutated KDM6B plasmids when 50% confluency is reached on the second day. Cells were collected and rinsed with ice-cold PBS 48 h after transfection. and then were lysed using radioimmunoprecipitation assay (RIPA) lysis buffer (ThermoFisher Scientific, Waltham, MA, USA) with cOmplete™ ULTRA Tablets (Sigma, St. Louis, MO, USA) for 20 min. Lysates were centrifuged for 20 min at 12,000 rpm at 4 °C. Supernatants were transferred to a new tube and protein concentration was determined using Pierce™ BCA Protein Assay Kit (ThermoFisher Scientific, Waltham, MA, USA). After being boiled in sample buffer for 5 min, proteins were loaded into wells (25 ug for each) of the 4–15% gradiant gel (Bio-Rad, Hercules, CA, USA) and run at 120 V for 90 min. Proteins were transferred from the gel to the Nitrocellulose membrane (Bio-Rad, Hercules, CA, USA) and blocked with 5% bovine serum albumin (BSA) in TBST for 1h. The membrane was incubated with Flag antibody (1:1000, ThermoFisher Scientific, MA1-91878), FOLR1 antibody (1:1000, Invitrogen, Waltham, MA, USA), H3K27me2 antibody (1:500, ThermoFisher Scientific, Waltham, MA, USA), H3K27Ac antibody (1:500, ThermoFisher Scientific, Waltham, MA, USA), Histone H3 antibody (1:5000, ThermoFisher Scientific, Waltham, MA, USA) and GAPDH antibody (Cell Signaling Technology, Danvers, MA, USA) overnight at 4 °C, and was incubated with HRP-conjugated anti-mouse antibody and anti-rabbit antibody (1:5000, Cell Signaling Technology, Danvers, MA, USA) for 1 h on the second day. The membrane was incubated with Enhanced Chemiluminescence (ECL) substrates (ThermoFisher Scientific, Waltham, MA, USA). Images were captured by BioRad ChemiDoc XRS Molecular Imager and Image J was used for image quantification.

### 2.8. Folate Receptor Alpha (FOLR1) Autoantibodies 

For each patient, FOLR1 blocking autoantibodies were measured in serum according to previously established methods [20]. Briefly, 200 µL of serum was acidified with 300 µL solution contains 0.1 M glycine, 0.5% Triton X-100 and 10 mM Ethylenediaminetetraacetic Acid (EDTA) and was added to 12.5 mg dextran-coated charcoal pellet to remove the free folate in the sample. After centrifugation, the supernatant was collected and neutralized with 40 µL 1 M dibasic sodium phosphate. The treated serum sample was incubated overnight with apo-FR purified from human milk. The next day, tritium [^3^H] folic acid (Moravek Inc., Brea, CA, USA, MT-783) was added and incubated for 20 min at room temperature. Free [^3^H] was removed by dextran-coated charcoal. The receptor-bound radioactivity in the supernatant fraction was determined. Blocking autoantibodies prevent the binding of [^3^H] folic acid to FOLR1. The autoantibodies titer was expressed as picomoles of FOLR1 blocked per milliliter of serum.

### 2.9. Statistical Analysis 

All data were analyzed via two-tailed Student’s *t*-test. Statistically significance is defined as *p*-values < 0.05. Data are presented as mean ± SEM.

## 3. Results

### 3.1. Missense Rare Variants in KDM6B Contribute to Human CFD

Exome sequencing was performed on 48 sporadic CFD patients’ DNA samples. No missense rare variants that were previously identified in reported candidate CFD genes (*FOLR1*, *PCFT*, *DHFR*, *CIC*) were observed. Using the filtering rules described above, we found 23 genes that have minor allele frequency >0.05 in our sporadic cases, and we considered them to be potentially enriched genes. After Bamcheck Validation and Fisher Exact Test, we determined that only one gene is highly enriched in this CFD cohort compared to the public dataset, and this gene is *KDM6B*. There were 7 allele counts in 96 alleles (7 in 48 cases), while, in over 250,000 gnomAD alleles, 5625 missense alleles are identified (*p* value = 0.0068). All six variants were predicted to be deleterious using SIFT software, indicating that they are likely pathogenic variants. Four of the six variants were highly (>80%) conserved across multiple mammalian species (Figure 1). Neither father nor mother DNA samples for the variant *KDM6B* p.Thr321Ala (c.A961G) were available for evaluation. The mother DNA sample was not available for the patient with variant KDM6B p.Ser826Tyr. For the other four *KDM6B* variants for which DNA samples were successfully collected from both parents, Sanger sequencing indicated that all five *KDM6B* variants were inherited from either their mother or their father (Appendix A). Sequencing data indicated that patient 1 carried biallelic *KDM6B* variants, with p.Thr761Ser inherited from the father and p.Arg1016Gln inherited from the mother.

### 3.2. Clinical Features of CFD Patients with KDM6B Variants

The clinical data are summarized in Table 1. All patients had severe psychomotor retardation with regression in two patients, as well as delay of speech development. Patients 1 and 5 were diagnosed with autism spectrum disorder. Dysmorphic features were observed in patient 4 and 5. The neurological picture evolving with advancing age was compatible with the clinical features described for the infantile-onset CFD syndrome, while all patients also presented with low CSF folate concentrations. CSF showed normal results for monoamine metabolites and pterins. Plasma homocysteine levels were in normal range for all five patients with *KDM6B* missense variants. Serum folate receptor-alpha autoantibodies were positive except for patient 3. In the siblings 2 and 3, a compound heterozygous (patient 2, EARS2: c.C322T, p.R108W, rs376103091 and c.1194C>G, p.Y398X, rs369291371) or heterozygous variant (patient 3, EARS2: c.C322T, p.R108W, rs376103091) of the *EARS2* gene was also identified, which may be partly responsible for the low CSF folate levels. Despite folinic acid treatment, all patients showed poor outcomes. Patient 1 experienced the most beneficial effects of the folinic acid therapy, and it is notable that treatment was initiated at the early age of 2 ½ months, clearly demonstrating the importance of early genetic testing and initiation of reduced folate therapy.

### 3.3. KDM6B Missense Variants Affect Protein Expression with No Effect on Localization

We performed functional analyses on these six *KDM6B* variants to determine the effects of *KDM6B* variants on protein expression and localization. KDM6B protein was expected to be located within the cell nucleus. As seen in Figure 2A, both wild-type and mutant flag-KDM6B were located within the nucleus which showed no effect of these variants on the cellular localization of KDM6B protein. Moreover, some of the KDM6B mutants showed a weaker signal and we speculated that the protein expression might be affected by *KDM6B* variants. Western blotting analysis was then performed, and compared to the wild-type, flag-KDM6B protein levels were significantly reduced in all the six variants (*p* < 0.01) (Figure 2B,C), which indicated that these *KDM6B* missense variants might decrease the KDM6B protein expression level or potentially affect the protein concentration.

### 3.4. KDM6B Missense Variants Downregulate FOLR1 Protein Level

FOLR1 has been shown to be an important receptor for the uptake of folates into the cells and play a crucial role in neural development. Previous genetic studies have demonstrated that human FOLR1 loss of function variants contribute to CFD [11]. We initially examined whether the *KDM6B* variants affected FOLR1 protein levels. As depicted in Figure 3A,B, wild-type flag-KDM6B increased FOLR1 protein level compared to control (*p* < 0.05), and overexpression of all the six flag-KDM6B mutants significantly decreased FOLR1 protein expression level compared to wild-type flag-KDM6B (*p* < 0.001).

### 3.5. KDM6B Missense Variants Upregulated H3K27me2 and Downregulated H3K27Ac

It is known that KDM6B plays a role in H3K27Me3/2 demethylation. Therefore, we investigated whether KDM6B variants affected its demethylation function. We initially examined the levels of H3K27me2 protein, which is the dominant modification form, in cells overexpressing wild-type and mutated KDM6B. As shown in Figure 4, H3K27me2 protein levels in wild-type KDM6B were significantly decreased compared to pcDNA3.1 control, while all six of the KDM6B variants significantly increased H3K27me2 abundance compared to wild-type KDM6B, demonstrating the loss of function or expression of KDM6B as lysine-specific demethylase. Histone H3 lysine 27 methylation and acetylation were reported to interact in an antagonistic manner [21]. As depicted in Figure 4, wild-type KDM6B significantly increased H3K27Ac protein levels compared to control, while all of the KDM6B mutant decreased H3K27Ac protein levels compared to wild-type KDM6B. It is obvious according to the quantification results that H3K27Ac protein levels in each group basically showed an opposite tendency with respect to H3K27me2 activity.

## 4. Discussion

In this study, we performed the first genetic association analysis of *KDM6B* and CFD, and six variants of *KDM6B* were identified among 48 sporadic CFD patients, all of which were predicted to be damaging. One of the patients was a biallelic *KDM6B* variant (p. p.Thr761Ser and p.Arg1016Gln) carrier with the lowest CSF folate concentrations among these five CFD patients, indicating a potential additive genetic model for *KDM6B* association with CFD. Two of the patients were siblings, both of whom inherited the *KDMB* p.Arg908Cys variant from their father. All variants identified in our CFD cohort were missense variants, and most of them were known to be inherited from parents, while those in Stolerman and colleagues’ neurodevelopment delay cohort were de novo nonsynonymous variants, including missense, frameshift insertion and stop gain variants. It is known that about 80% of CFD patients suffer from neurodevelopment delays. In this study, all five patients with *KDM6B* variants had neurodevelopment problems, including autism, language delay and intellectual disability. It is unknown whether the patients in Stolerman and colleagues’ KDM6B study had CFD. Based on Stolerman and our KDM6B study, *KDM6B* loss of function variants can cause neurodevelopment defects. Although some parents were heterozygous for KDM6B missense variants, they lacked any CFD-associated phenotypes; thus, we assume that single rare deleterious heterozygous *KDM6B* missense variants may not be sufficient to cause CFD. That said, such variants interacting with other genetic (e.g., Patient 3 KDM6B p.R908C and EARS2 p.R108W) or/and environment factors, as well as biallelic KDM6B variants (e.g., patient 1 KDM6B p.T761S and KDM6B p.R1016Q), might be sufficient to induce the CFD syndrome phenotype.

We also performed further experiments to explore the possible underlying mechanisms by which KDM6B contributes to the etiology of CFD. The most common primary causes of CFD include dysfunction of FOLR1 and elevated serum FOLR1 autoantibody titers [22,23]. Along with Slc46a1, FOLR1 is highly expressed in the choroid plexus [24] and mediates the transportation of folate across the choroid plexus into the cerebrospinal fluid. Decreased FOLR1 leads to a folate deficiency in CSF, and the delayed CFD onset is usually considered to be the result of preserved 5-MTHF transport by FOLR2 compensating for FOLR1 deficiency in early development [1]. In this study, no variants of *FOLR1* or *FOLR2* were identified in patients with *KDM6B* variants. Through in vitro experiments, we found that, consistent with KDM6B protein levels, FOLR1 protein levels in HeLa cells overexpressing wild-type KDM6B were increased compared to the control, while FOLR1 levels in all six of the *KDM6B* variants were significantly decreased compared to the wild-type KDM6B.

The methylation of H3K27 is frequently associated with gene repression [25] and plays an important role in embryonic stem cell (ESC) self-renewal, differentiation, and dynamic development [15,26,27]. In this study, we also observed that dysfunction or decreased KDM6B protein resulted in increased H3K27me2, and, accordingly, H3K27Ac showed opposite tendency of protein abundance as H3K27me2, which is consistent with previous publications. H3K27ac is well recognized as a marker for active enhancers and promoters that is strongly correlated with gene expression and transcription-factor binding [28]. Sarah et al. performed a histone acetylome-wide association study and identified disease-associated H3K27ac differences in the entorhinal cortex from Alzheimer’s disease patients [29]. Previous studies have also demonstrated that folic acid and folate receptors (FOLRs) play an important role in the downregulation of homocysteine, a risk factor of Alzheimer’s disease [30], and, together with our results, we suspect that the decreased FOLR1 protein levels in *KDM6B* variants might be due to decreased H3K27Ac which might compromise FOLR1 transcription (Appendix A).

We found that 80% (4/5) of these *KDM6B* variant heterozygous CFD patients had high titers of FOLR1 autoantibodies, which have been widely believed to contribute to the occurrence of CFD. It is hard to elucidate the relative contribution of FOLR1 autoantibodies and KDM6B variants on the actual etiology of CFD in this study. However, according to the literature, Histone H3 lysine 27 demethylases (KDM6B, KDM6A) were reported to be required for T-cell differentiation through regulating Jmjd3-Irf4 axis [31,32,33]. KDM6B was found to play important roles in autoimmune reactions in recent studies. Using Kdm6b-deficient mice, Liu and colleagues found that Kdm6b is essential to maintain the postnatal thymic medulla homeostasis promoting medullary thymic epithelial cells (mTECs) survival and regulating the expression of tissue-restricted antigen (TRA) genes. BALB/c nude mice transplanted with Kdm6b-/- thymic presented inflammatory infiltrates in several tissues, which directly demonstrated that the Kdmb6 deficiency could induce an autoimmune reaction [34]. Therefore, we suspected that, in our study, KDM6B variants induced FOLR1 autoantibodies, which, in turn, contributed to the CFD phenotype together with the reduced FOLR1 expression level induced by the KDM6B variants. However, this hypothesis needs to be more rigorously tested in the future.

We also investigated the remainder of the cohort for autoantibodies and found that more than 89% of CFD patients tested positive for autoantibodies [22]. The common shared phenotype was the low folate concentrations in the CSF. The other common symptoms included autism, intellectual disability, and language delay. Because neural development and autoimmunity could be regulated by thousands of genes, and each individual may have different genetics variants, which would affect some aspect of neural development, it is not surprising that clinical signs differed from patient to patient. We are working on other genes that may contribute to both FOLR1 autoantibodies and CFD phenotypes.

To our knowledge, this is the first report of KDM6B variants being associated with human CFD syndrome, and our functional analyses indicated that CFD associated *KDM6B* variants affected FOLR1 protein levels. H3K27 methylation modification levels were also disrupted by CFD associated *KDM6B* variants. Future investigations are necessary to explore the underlying pathophysiological mechanisms in more detail, to enhance our understanding of the etiology of CFD and provide a theoretical basis for the treatment of these devastating neurodevelopmental disorders.

## 5. Conclusions

In this study, we identified 6 *KDM6B* variants from 48 isolated CFD cases. We also performed several functional experiments and determined that *KDM6B* variants decreased the amount of KDM6B protein, which resulted in elevated H3K27me2, lower H3K27Ac and decreased FOLR1 protein concentrations. In addition, FOLR1 autoantibodies have been identified in patients’ serum. To the best of our knowledge, this is the first study indicating that *KDM6B* may be a novel CFD candidate gene in humans. Our findings will enhance our understanding of a new CFD etiology.

## Figures and Tables

**Figure 1 biology-12-00074-f001:**
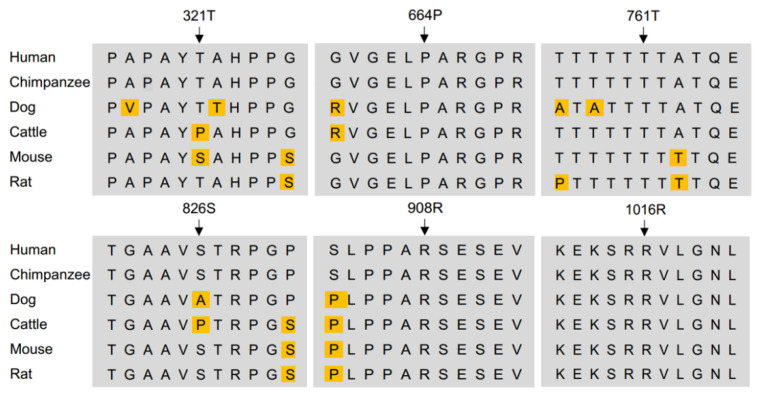
Conservation of amino acid of identified KDM6B variants among species. KDM6B protein sequence of human, chimpanzee, dog, cattle, mouse, and rat were use in the blast.

**Figure 2 biology-12-00074-f002:**
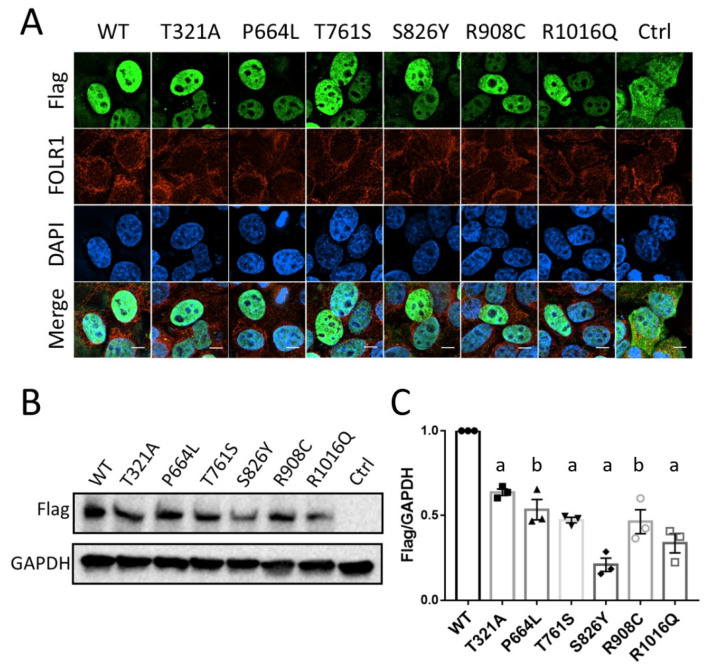
Subcellular localization and protein abundance of KDM6B wildtype and KDM6B mutant. (**A**) HeLa cells were transfected with mutated and wild-type constructs of Flag-tagged KDM6B and pcDNA3.1 backbone vector for 36 h and were imaged under deconvolution microscope. Scale bar 5 um. (**B**) Western blotting was performed in HeLa cells 48 h post transfection. GAPDH was used as loading control. (**C**) Western blotting was repeated three times and a Student *t*-test was performed to compare the protein level between wild-type and mutant. Significance compared to WT: “a” represents *p* < 0.001; “b” represents *p* < 0.01.

**Figure 3 biology-12-00074-f003:**
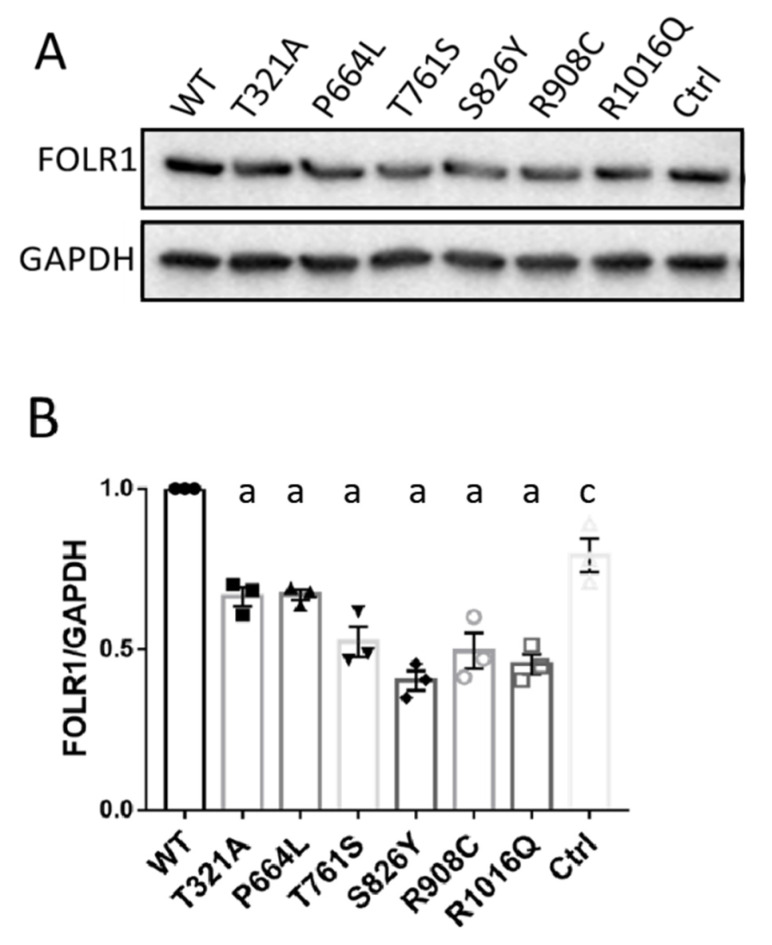
Overexpression of KDM6B mutants affected FOLR1 protein level and folate uptake ability of HeLa cells. (**A**) HeLa cells were transfected with mutated and wild-type constructs of Flag-tagged KDM6B and pcDNA3.1 backbone vector for 48 h, and Western blotting was performed to quantify FOLR1 protein level in each group. GAPDH was used as loading control. (**B**) Western blotting was repeated three times and a Student *t*-test was performed to compare the protein levels between wild-type and mutant. Significance compared to WT: “a” represents *p* < 0.001; “c” represents *p* < 0.05.

**Figure 4 biology-12-00074-f004:**
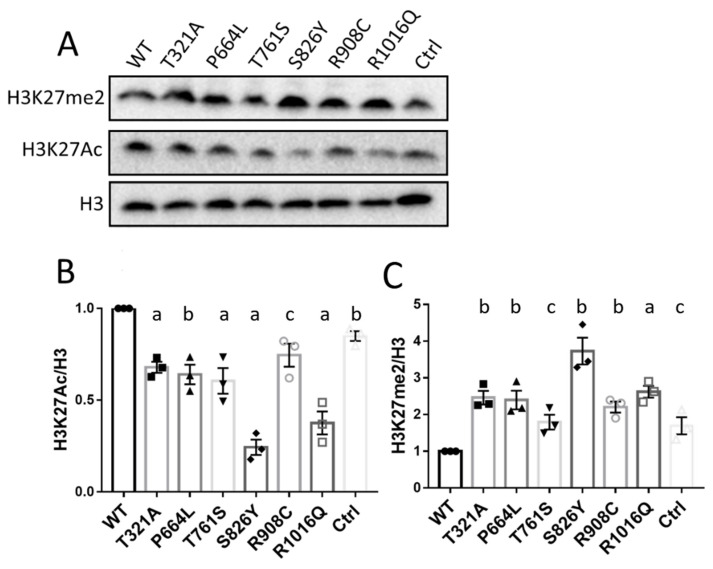
Overexpression of KDM6B mutants affected H3K27 demethylation and acetylation in HeLa cells. (**A**) HeLa cells were transfected with mutated and wild-type constructs of Flag-tagged KDM6B and pcDNA3.1 backbone vector for 48 h, and Western blotting was performed to quantify H3K27me2 and H3K27Ac protein levels in each group. H3 was used as loading control. (**B**,**C**) Western blotting was repeated three times and a Student *t*-test was performed to compare protein levels between wild-type and mutant of H3K27me2 and H3K27Ac, respectively. Significance compared to WT: “a” represents *p* < 0.001; “b” represents *p* < 0.01; “c” represents *p* < 0.05.

**Table 1 biology-12-00074-t001:** Clinical features of patients with *KDM6B* variants.

Features	Patient 1 *	Patient 2	Patient 3 **	Patient 4 ***	Patient 5
Gender	F	M	F	F	F
Age (years at evaluation)	¼	5 ½	4	11 ½	11
*KDM6B* gene variant	c.C2282G p.T761S; c.G3047A p.R1016Q	c.C2722T p.R908C	c.C2722T p.R908C	c.C2477A p.S826Y	c.C1991T p.P664L
Pregnancy	Epilepsy, treated with VPA ^b^ + FA ^c^	normal	normal	N/A ^a^	Pre-eclampsia
Growth parameters Height Weight Head circumference	75–90th 75th 50–70th	3rd 3rd 3rd	25th 25th <3rd	3rd <3rd <3rd	N/A N/A 50th
Dysmorphic features	none	none	none	Coarse facial features	Facial; Pulmonary valve stenosis
Neurological features ° Unrest, insomnia ° Decelerating head growth ° Psychomotor delay and Regression ° Hypotonia and ataxia ° Pyramidal dysfunction ° Dyskinesias (chorea, athetosis) ° Epilepsy	+ - + - + - - -	+ + + + + + - +	- + - + - - + At 9 months ** Epileptic status and liver failure	+ + + - - - - +	+ N/A + + - - - +
Cognitive functions ° Language delays ° Intellectual disability	- -	+ +	+ +	+ +	+ +
Autism spectrum disorder	+	-	-	-	+
Neuro-imaging	Normal	LTBL ^d^; partial recovery of white matter changes	Progressive cortical/cerebellar atrophy	Subcortical white matter lesions	N/A
Spinal fluid folate (nmol/L)	14	24	38	30	34
% of bottom reference CSF	22.2%	58.5%	60.3%	73.2%	83%
Serum FRα antibodies	+	+	-	+	+
Start folinic acid treatment	2 ½ months	5 years	3 ½ years	11 ½ months	None

* Patient 1. In the patient, her brother and mother a mutation of the NFκB1 gene was found. ** Patient 3 (younger sister of patient 2) had normal development until 9 months, but then suffered from severe epileptic status with transient liver failure, hypoglycemia and lactic acidosis. Seizures have been treated with clonazepam, valproic acid and phenobarbitone. Since this episode she developed irritability, had a deceleration of head growth, choreatic movements and cognitive decline. MRI showed progressive cerebellar and cortical atrophy. Alpers disease could be excluded with normal POLG1 gene findings. *** Patient 4 was found to carry a duplication at chromosome 7q31.32-q32.3. ° CSF 5-methyltetrahydrofolate levels were low in all patients. Reference range for healthy controls aged between 0–4 years: 63–111 nmol/L; age 5–16 years: 41–117 nmol/L. Patient 1–4 received folinic acid treatment, whereas a repeat spinal tap for patient 5 showed normal CSF folate. Abbreviations: ^a^ N/A data not available. ^b^ VPA valproic acid. ^c^ FA folic acid. ^d^ LTBL leukoencephalopathy with thalamus and brainstem involvement and high lactate.

## Data Availability

Data pertaining to specific variants generated during the downstream analyses, which support the findings of this study, are available upon request to the corresponding author (R.H.F.). De-identified data will be made available upon request.

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
