# Peer review of "KDM6B Variants May Contribute to the Pathophysiology of Human Cerebral Folate Deficiency"

_biology, 2022, doi:10.3390/biology12010074_

Round 1

Reviewer 1 Report

This is an interesting paper describing an association of genetic variants in the demethylase gene KDM6B with a change in FOLR1 expression and cerebral folate deficiency.

From the total cohort it is interesting though not clear why the authors focus on KDM6B when the cases involving this gene described previously had such diverse set of multiple issues, not just neural, by contrast to most described CFD cases? And this is more unclear when only 5 of their 48 cases had variants in this gene but they also had the autoantibodies characteristic of this condition.

Furthermore, the authors do not comment on the relative contribution of FOLR1 autoantibodies and KDM6B variants on the CFD. Given that the majority of the cohort did not have a KDM6B variant, some discussion of this would be valuable.

Did the authors investigate the rest of the cohort for autoantibodies and compare clinical signs and symptoms between these and the KDM6B variants? That would also be interesting to compare as the KDM6B may turn out to be a minor contributor in vivo, where autoantibodies may be the major driver in CFD syndromes. This would be a much more impactful study if this was included.

The authors make a statement in their abstract that decreased FOLR1 may predispose people to autoantibodies but this proposal is not mentioned or discussed further.

Reviewer 2 Report

This is really well written paper. Experiments are conducted properly. Those are really interesting data and i believe paper might be cited.

Author Response

We would like to thank you for taking the time and effort necessary to review the manuscript. We sincerely appreciated your affirmation and encouragement.
